# Influencing Factors, Mechanism and Prevention of Construction Workers’ Unsafe Behaviors: A Systematic Literature Review

**DOI:** 10.3390/ijerph18052644

**Published:** 2021-03-05

**Authors:** Qingfeng Meng, Wenyao Liu, Zhen Li, Xin Hu

**Affiliations:** 1School of Management, Jiangsu University, 301 Xuefu Road, Zhenjiang 212013, China; mqf@ujs.edu.cn (Q.M.); 2221910009@stmail.ujs.edu.cn (W.L.); 2School of Architecture and Built Environment, Deakin University, 1 Gheringhap Street, Geelong, VIC 3220, Australia; xin.hu@deakin.edu.au

**Keywords:** construction workers, unsafe behaviors, influencing factors, formation mechanism, pre-control methods

## Abstract

Unsafe behaviors of construction workers are one of the main causes of accidents at construction sites. The research on unsafe behaviors of workers helps to reduce the incidence of accidents and has attracted much attention. However, a systematic literature review in this field is still lacking, which hinders stakeholders’ comprehensive understanding of the unsafe behaviors of construction workers. Therefore, the aim of this study is to address this research gap based on retrieved literature from the Web of Science. First, the study conducted a descriptive analysis of the year, quantity, publishing organization, and keywords of the literature. In addition, three research topics were identified and discussed, including the influencing factors of construction workers’ unsafe behaviors, the formation mechanism of unsafe behaviors, and the pre-control methods of unsafe behaviors. Moreover, a research framework was proposed and future research directions were also suggested. The research findings promote stakeholders’ understanding of the influencing factors, formation mechanism, and pre-control methods of construction workers’ unsafe behaviors, and lead to future research directions in the studied field.

## 1. Introduction

The construction industry is a crucial sector of the national economy. Nevertheless, its development has been plagued by frequent accidents and injuries [1,2,3,4]. It is estimated that about 60,000 people die in construction accidents in the world every year, which is equivalent to one accident every 9 minutes [1]. About 80% of construction accidents were caused by workers’ unsafe behaviors [5,6,7]. Given that workers’ unsafe behaviors are the root cause of construction accidents, it is of great importance to propose appropriate managerial strategies to deal with workers’ unsafe behaviors in practice [8]. This has also contributed to the considerable explorations about unsafe behaviors of construction workers [9].

The unsafe behaviors of construction workers refer to their dangerous practices in violation of organizational discipline, operating procedures and methods in their professional activities [8]. The unsafe behaviors of construction workers are a relatively complex phenomenon, which is often related to many factors [10]. This has prompted scholars’ explorations about the factors impacting unsafe behaviors of construction workers from different perspectives, including “individual”, “organizational management”, “project”, and external “production operations”. Scholars also explored a set of crucial research questions such as if construction safety management should be taken as a system, how different factors affect construction workers, and how unsafe behaviors further affect this system [11]? For example, Al-Bayati et al. [12] think that when organizational management factors are the root cause, the cause of the accident may be a system error, not human error. In addition, the concept of multiple causalities introduced by Petersen [13] has given great enlightenment to the research in the field. According to multiple causality, multiple influencing factors are combined in a random manner, which causes the occurrence of unsafe behaviors and accidents at construction sites. Applying this concept to the construction industry helps to define and explain the mechanism of unsafe behaviors of construction workers. Scholars also used different research theories and methods to explore the formation mechanism of workers’ unsafe behaviors such as the Structural equation model and system dynamics. At last, given the importance of taking measures to promote safe behaviors of construction workers, explorations regarding pre-control measures were proposed from different perspectives such as technology and management perspectives.

Although there can be found an increased number of explorations on unsafe behaviors of construction workers, a relatively complete system framework is still lacking.The so-called framework is a process to study the unsafe behaviors of construction workers. To address this research gap, the study conducted a literature review by using the method of system analysis. The review work focuses on the three crucial topics of “influencing factors”, “formation mechanism” and “pre-control methods” of unsafe behaviors of construction workers. On the basis, a detailed framework was proposed, and future research directions were suggested. It is expected that the research findings will promote stakeholders’ comprehensive of understanding about unsafe behaviors of construction workers and safety management of construction site, and also hopes to point out the future research direction for researchers in this field.

The article is divided into five sections. Section 2 detailly explains the research methods and literature analysis process in this study. Section 3 provides the state-of-the-art review of related literature, which was discussed from the three perspectives of influencing factors, formation mechanism, and pre-control methods of unsafe behaviors of construction workers. Section 4 suggests the future research directions based on the content analysis results of the literature. Finally, Section 5 clarifies the conclusion of this article.

## 2. Literature Sources and Analysis

### 2.1. Literature Source

This article uses the “Web of Science” database as the source of literature search. Through a large amount of literature reading, it is found that some articles do not involve the word “construction” in the theme, and the content is related to the unsafe behavior of construction workers, therefore, in order to ensure a more comprehensive literature search, the search topic was expanded from topic 1 “unsafe behavior of construction workers” to topic 2 “unsafe behavior of workers”. Search is limited to the “core collection of science net” because these articles are more authoritative and representative in related fields. Additionally, the time range of the literature searching was set as 1986–2020 to receive a comprehensive research. The searching results showed 147 articles related to topic 1 “unsafe behavior of construction workers” and 380 articles related to topic 2 “unsafe behavior of workers” at the time of searching (20 May 2020). It was found that the searched documents of topic 1 are all included in the documents of topic 2. After detailed carding, an article screening process was proposed to identify papers that can be reviewed in this study (Figure 1). Finally, the screening process led to the identification of 140 related examples of literature. 

### 2.2. Literature Analysis

#### 2.2.1. Literature Publication Year and Quantity Trend Analysis

Figure 2 shows the number of publications in different years. It can be seen that few studies on the “unsafe behavior of construction workers” were published before 2010. Nevertheless, the number of publications has increased gradually since 2010, with the largest publication number arriving at 31 in 2018. This reflects the increased research interest in exploring the topic of “behaviors of construction workers” in recent years. 

#### 2.2.2. Literature Publication Source Analysis

Figure 3 intuitively presents the publication sources by using the visual analysis tool Citespace. A circle represents a journal. The larger the circle is, the more papers published in this journal. It was found that most papers were published in the fields of “risk and safety management” (e.g., Safety Science, Journal of Safety Research) and “engineering management” (e.g., Journal of Construction Engineering and Management, Automation in Construction). 

#### 2.2.3. Keyword Cluster Analysis

Keyword cluster analysis was also conducted in this study to identify and analyze the key co-words appearing in many articles. These co-words with higher frequency are often the hot topics in a field. Consequently, the analysis can be used to identify the sub-topics heavily explored in the “unsafe behaviors of construction workers” field. In total, 180 keywords with their usage frequency being no less than 3 were identified in this study. These keywords were primarily classified into nine sets by using the Force Atlas algorithm(an algorithm that can show the importance and relationship of nodes). A further investigation found that some of them can be integrated based on their similar meanings (As shown in the Figure 4, the purple circle "safety" and the brown circle "construction safety" can be combined into one type). Finally, a total of six sets were obtained (Table 1). Figure 4 shows the relationship between keywords and sets by using VOSviewer. Different colors represent different sets. Additionally, the size of a node indicates its frequency of being cited. Moreover, the distance between two nodes indicates the strength of their relationship (a longer distance means a weaker relationship). As shown in Figure 4, the central keywords include “Safety Climate”, “Performance”, “Management”, “Injury”, and "Construction Safety"(the largest circle of each color). These keywords reflects results (e.g., forming a safety atmosphere, causing some behavior or injury), whereas their surrounding keywords are factors and measures contributing to the results. This also further proves that the research direction of this article is to analyze the content of the collected literature in detail from the three perspectives of influencing factors, the interaction between factors and pre-control measures.

## 3. A State-of-the-Art Review of Related Literature

Hoyos [14] believed that it is important and necessary to conduct in-depth research on the causes of construction accidents as this helps propose preventive measures used to improve the safety performance of construction projects. A close examination of the historical studies in the “unsafe behaviors of construction workers” field found that three sub-topics were heavily explored historically, including the factors affecting workers’ unsafe behaviors, the formation mechanism of construction workers’ unsafe behaviors, and pre-control measures of construction accidents. On the basis, a research framework in the studied field was proposed (Figure 5).

### 3.1. Factors Affecting Construction Workers’ Unsafe Behaviors

The factors affecting the unsafe behaviors of construction workers have been explored heavily. Based on in-depth semi-structured interviews with professionals from Hong Kong, Choudhry et al. [10] found that these factors are associated with management, safety procedures, psychological characteristics, economic pressure, self-esteem, experience, performance pressure, perceived risk, work environment, and safety education and training. By using an effective conceptual framework, Zid et al. [15] divided these factors into the three levels, namely organizational factors, safety climate factors, and individual factors. On this basis, this paper makes a more detailed division of influencing factors from the individual level, organization management level, project level and production operation level.

#### 3.1.1. Individual Factor Level

This level focuses on individual construction workers, such as their physiology, psychology, personality characteristics, Subjective attitude consciousness, risk perception and cultural differences. Table 2 lists the identified individual factors and their supporting references.

When it comes to the physiological factors of workers, the first consideration is the physical condition of the workers. Construction workers are faced with various health problems, including physical fatigue, cardiovascular and cerebrovascular diseases, muscle and bone diseases, and high pressure in life and work, which increase the occurrence of unsafe behaviors at work [16]. Based on the Construction Worker Fatigue Rating Scale (FASCW), Fang et al. [17] used the level of fatigue to measure the safety performance of workersand found that workers make more mistakes when they are fatigued. Syamlal et al. [18] found that construction workers are the group with the largest number of smokers when compared with those from other sectors. It is not only easy to cause lung infection, cough and health problems, but also easily distracts during smoking and leads to accidents.

Psychological factors can drive people’s behaviors. Yang and Byung [16] confirmed that the occupational pressure and social psychological pressure of construction workers have greater impacts than that of physical fatigue. Leung et al. [19] classified the pressure that affects the safety behaviors of construction workers into the two types of work pressure and emotional pressure and found that emotional pressure is more likely to affect workers’ behaviors. Kim [20] also recognized the importance of emotions to safe behaviors, suggested that construction sites should create a good perceptual safety culture to create perceptual safety environments. Ju et al. [21] stated that emotional exhaustion can lead to the occurrence of unsafe behaviors, and good group safety regulations can help reduce the triggering effect of emotions on workers’ unsafe behaviors and help the psychological adjustment. Wang et al. [22] showed that psychological adjustment can alleviate related safety pressures, increase worker participation, and improve unsafe behaviors. Chen and Li [23] pointed out that the influences of some external factors (e.g., Construction environment) on workers’ unsafe behaviors are linked to the mediating variable "emotion". On the basis of determining six dimensions of work stress, Wu et al. [24] found that work stress is negatively correlated with safety behaviors through inspection. By using a system dynamics (SD) model, Mohammadi and Tavakolan [2] proved that work pressure can also have negative impacts on workers’ safety behaviors and safety performance.

Historical studies also confirmed that personality characteristics also affect workers’ safety behaviors. Based on the Eysenck Personality Questionnaire, Sing et al. [25] found that there is a correlation between the safety behaviors of construction workers and their personality. For example, people with higher scores in the dimension of psychotic personality tend to be narrow-minded and more likely to have accidents on construction sites [25]. By using structural equation modeling (SEM), Chen et al. [26] verified that different personalities of workers can lead to interpersonal conflicts, which results in the occurrence of unsafe behaviors on site. On the contrary, workers with better personality and better individual adjustment ability generally do not lead to conflicts, and their unsafe behaviors are relatively less. Hasanzadeh et al. [27] believed that the personality characteristics of workers will affect their attention and situational awareness in the workplace. For example, careless workers are prone to inattention, leading to unsafe behaviors [27]. Hasanzadeh et al. [3] further explained the relationships between personality characteristics and attention and risk identification, and verified that the personality dimensions of workers, especially extroversion, conscientiousness and experience, have significant influences on their attention allocation. By using the theory of planned behavior (TPB), Zhang et al. [4] verified the relationship between workers’ personality characteristics and attention and risk recognition, and explained the guiding role of personality characteristics for behaviors.

Attitude determines behaviors. Cavazza and Serpe [28] started from the ambivalence of whether construction workers wear personal protective equipment (PPE), and verified that construction workers with low levels of PPE use are more likely to have unsafe behaviors and cause accidents. Through combining the theory of planned behavior and SEM, Xu et al. [29] found the influences of workers’ attitudes and ambivalence on behavioral intentions. Gyu-sun et al. [30] clarified that unfounded responsibility transfer and prevarication among workers from accident responsibility awareness and attitudes will further aggravate the occurrence of unsafe behaviors.

Workers can perceive risks in time, which is important for them to make timely judgments and avoid risks. Burns and Conchie [31] analyzed workers’ risk perception source preferences from the five major management occupations and eight major risk sources in the construction industry, and explored where workers tend to obtain risk information. By using SEM, Huang et al. [32] verified that workers’ risk perception and unsafe behavior are negatively correlated, that is, the stronger the risk perception ability, the fewer unsafe behaviors. Man et al. [33] divided workers’ risk perception into three dimensions and produced a Risk Perception Scale (CoWoRP) to measure the risk perception ability of construction workers to determine that construction workers are most likely to perceive under hazardous work conditions to lower risk characteristics.

Due to regional and cultural differences, communication between workers (especially from different countries and regions) will have language barriers. Al-Bayati et al. [34] believe that language and cultural barriers are the main sources of communication failure, and some unclear construction instructions will lead to human error and increase the incidence of accidents. In addition, some language communication barriers can also cause a certain degree of psychological pressure. This communication pressure makes some workers unwilling to report their work problems to their foreign supervisors in time, which also leads to some unsafe behaviors repeatedly [34]. In subsequent research, Al-Bayati et al. [35] added the unit of "cultural diversity training" to the safety training module on construction sites in the United States. After providing Hispanic workers with Spanish scene training, workers’ safety capabilities were greatly improved. This case also proves the importance of language and cultural communication. Chan et al. [36] also suggested that the construction industry in developed countries should try to train migrant workers in their native language when training them; at the same time, they also encourage migrant workers to learn the local language. Through these methods, it is helpful for safe communication between workers and interaction between workers and supervisors, and the incidence of unsafe behaviors will also be greatly reduced.

#### 3.1.2. Organizational Management Factor Level

This level includes the two aspects of organization and management. Organizational factors mainly include safety climate and group norms. The influences of managers are more manifested in their leadership and management methods. Table 3 lists the factors at the organization and management level.

D. Zohar et al. [53] believe that safety climate is the concern of employees on safety, and this kind of concern belongs to the perception of holism. Nicole et al. [54] proposed that the two major factors of safety climate in the construction industry should be the obligation of management (e.g., safety commitment, safety compliance, safety training) and workers’ safety participation. In 2008, NCA defined safety culture as “basic organizational principles, norms, commitments and values related to how safety and health work and their relative importance relative to other workplace goals” [55]. This is almost consistent with the obligation of management [54]. In other words, the obligation of management is to create a good enterprise safety culture, so as to enhance safety climate level. Zhou et al. [37] believed that management’s safety commitment is the core of safety climate, and good safety commitment helps to improve the safety climate and improve workers’ behavior. Fargnoli and Lombardi [38] believe that the management should start with human behavior and formulate a series of reasonable safety rules and regulations, so as to create a good safety climate. He et al. [39] verified by SEM that safety climate is positively correlated with safety compliance and safety participation behavior, and good safety compliance behavior and workers’ safety participation contribute to the improvement of safety climate. Fang et al. [40] also considered that workers’ safety participation is an important part of a good safety climate.

The group norms of workers will also have profound impacts on the behaviors of individual workers. Arcury et al. [41] used an interactive voice response (IVR) system to collect the communication information of construction workers, and found that workers’ collective safety practices are positively correlated with the safety climate. Choi and Lee [42] proposed that active group norms are an effective means to improve workers’ safety behavior. In their subsequent research, Choi and Lee [43] established an agent-based model to explain workers’ cognitive abilities and group norms. Based on correlation analysis, regression analysis, and t-test, Choi et al. [44] used behavioral economic experiments to verify that personal standards of safety behavior of construction workers are significantly affected by their perceived group norms, the behavior of construction workers is largely consistent with the behavior of their group.

Construction managers should have an exemplary role and considerable leadership ability to convince their workers. From the perspective of managers’ influence on workers, Fang et al. [40] verified that the supervision behavior of managers plays an important roles in improving the safety performance of construction workers. Shen et al. [45] found that transformational leadership (e.g., caring for workers, encouraging workers) is more conducive to creating a good safety climate and improving workers’ safety capabilities than traditional leadership methods (e.g., monitoring, ordering). Xiong et al. [46] stated that it is important for workers to have an opinion-led leader as this can guide workers to develop the correct safety awareness and also highlights the importance of leadership.

Reasonable management methods can make the interior of the construction site orderly. When solving complex construction problems, Du et al. [47] proposed to screen and compare management methods by using non-structural fuzzy decision-making methods to rank various management methods and select the most suitable method. Sheng et al. [48] used the fault tree analysis method to analyze the internal problems of the construction organization and proposed a series of management methods for the problem. Choudhry [49] pointed out that management methods should be selected mainly based on "behavior" to create the "behavior-based security (BBS)". On this basis, Li et al. [9] proposed an extension method, namely "Proactive Behavioral Safety (PBBS)". Cavazza and Serpe [50], Hai and Zhu [51], and Harsini et al. [52] all believed that the use of safety education and training methods can improve workers’ unsafe behaviors.

#### 3.1.3. Influencing Factors at Project Level

The completion of an engineering project requires the cooperation of multiple organizations. In the construction process, there is an interactive relationship between the construction subject (construction unit) and many stakeholders (shown in Figure 6). 

The whole process has a clear hierarchy. Through the commission of the upstream organization and the orderly cooperation of the downstream organization, the project can be completed smoothly.Because the construction subject interacts with multiple organizations, this also makes the safety behavior of workers in the construction subject to a certain extent dependent on other organizations. Therefore, from the perspective of the project as a whole, security issues between organizations should also be discussed, and security issues involving organizations need to start with the hierarchy and interactivity between organizations. Table 4 lists two influencing factors at the project level, namely safety investment and safety inspection feedback.

Safety investment is the most basic guarantee for safety production. Due to the hierarchical safety of construction projects, insufficient funds and large arrears of safety investment, the coordinated operation of each level will be affected. Kim and Park [56] believes that the inadequate investment of workers’ safety equipment and safety facilities in the workplace is the main cause of safety accidents. Due to the lack of safety investment, the shortage of funds for safety education and training of workers, the level of safety knowledge can not be improved, which also increases the incidence of safety accidents [40].

Due to the hierarchical nature of engineering project organizations, downstream organizations are often subject to the supervision of upstream organizations. The higher-level project management agency will regularly send personnel to the lower-level construction units to conduct safety inspections. In this process, the upper-level management agency will further popularize safety for the lower-level construction units Policies and standards [57,58], analysis and prevention of existing safety hazards [59,60], while the upper management organization will also listen to the feedback information of the construction unit [61], so as to conduct a comprehensive safety management system audit [62]. The interaction of managers between different organizations also contributes to the exchange of management experience [61], thereby better promoting the order and safety of the construction site. This kind of communication and interaction between organizations starting from the project as a whole has greatly improved the safety performance of the entire project.

#### 3.1.4. Production and Operation Factor Level

Safety accidents are most likely to occur in the process of production operations, and the unsafe behavior factors that lead to accidents are also the most intuitively visible in the process. This is closely related to the operation methods, operating environment, and construction equipment. Table 5 shows the identified production and operation factors.

Johnson et al. [63] pointed out that the unsafe behaviors of working at height are one of the main causes of unsafe accidents. Kaskutas et al. [64] also found that falling from height is a direct factor causing workers’ casualties, and confirmed the relationships between some other factors (e.g., Inattention) and falling from height through confirmatory factor analysis and multiple regression analysis. In addition, Fang et al. [65], Kolar et al. [66], Yin et al. [67], Shokouhi et al. [68], and Shi et al. [69] all confirmed the danger of working at heights, and these studies also proposed relevant pre-control measures (see Section 3.3 pre-control measures). Niu and Chen [70] used the redefined hazard and operability study (HAZOP) to identify possible unsafe behaviors of workers during lifting. Eskisar et al. [71] discussed piling accidents caused by workers’ unsafe behaviors.

Chi et al. [72] proposed that when workers’ unsafe behaviors (e.g., misjudgment or improper operation) are combined with unsafe working conditions (e.g., working conditions or climate), they become the main source of construction accidents. The working environment also has great influences on the occurrence of some accidents, mainly including the natural environment and the human environment [72]. Jiang et al. [11] believe that exposure to the open air is the main environmental factor that leads to unsafe behaviors of workers. Lu and Davis [73] studied the impacts of construction noise on workers and their judgment and decision-making on safety behaviors. Chen et al. [74] stated that construction safety signs can help improve workers’ awareness and warn workers to make safe decisions. These signs can also help create a good working environment [74]. Fang et al. [40] pointed out that a reasonable regulatory environment created by construction managers helps workers to participate in safety and improve safety behaviors. Both Mohamed et al. [75] and Del Puerto et al. [76] found that differences in different ethnic cultures and social environments will cause differences in workers’ perceptions, which leads to unsafe behaviors.

Workers mainly use two types of equipment in construction, namely operating equipment and protective equipment. The inappropriate use of this equipment will cause dangers. For instance, Zhao et al. [77] found that improper use of electrical equipment by construction workers will result in a high risk of electric shock and cause electrical safety accidents. Niu and Chen [70] pointed out that the correct and reasonable operation of site equipment can reduce the risk of accidents. Kaskutas et al. [78] exemplified the use of a series of protective equipment such as ladders, scaffolding, safety ropes, and gloves for the purpose of reducing the risk of falling from a height. Li et al. [79] stated that wearing safety helmets correctly can reduce the incidence of accidents. Zhang et al. [80] pointed out that collisions between workers and equipment will lead to unsafe accidents.

Regarding the influencing factors, there are many articles based on the research of individual factors of workers. Additionally, and many of the contents of organizational management factors and production operation factors are also closely related to the cognition and behavior of individual workers. Therefore, individual factors are the main factors, through the influence of individual factors can drive the influence of the other two factors. The project-level influencing factors are closely related to the interaction between organizations.

### 3.2. Formation Mechanism of Construction Workers’ Unsafe Behaviors

Mechanism refers to the operating rules and principles of the interconnection and interaction of various elements in a system [13]. When scholars explore the influencing factors of construction workers, they should also explore how these factors affect the unsafe behaviors of construction workers, which factor is the key influencing factor, that is, further deepen the analysis of influencing factors. This is the formation mechanism of unsafe behaviors of construction workers. In order to better find the key causes of unsafe behavior, scholars take the workers as the core subject, and constantly combine a series of influencing factors around the workers to explore, find the factors with greater influence effect and make improvement. As the research content is diverse and involves many factors at different levels, the occurrence mechanism is divided and analyzed from the perspective of the research method.

**Structural equation modeling (SEM):** In the field of safety, it is the most common to use SEM to analyze the relationship between variables (factors). Khosravi et al. [81] constructed SEM from the perspective of safety supervisors for workers’ individual factors, safety atmosphere and environmental conditions in the workplace, and concluded that the safety status of the workplace plays crucial roles in linking the safety atmosphere and worker participation. Goh and Sa’adon [82] used the Planned Behavior Theory (TPB) to model the cognitive factors that affect the behavior of workers at heights, and studied them and determined which TPB structure (attitude, subjective norms, perceived behavior control and intention) is potentially important influencing factors of safety behaviors at heights. Guo et al. [83] developed and tested eight competition models related to the safety behaviors of construction workers to better understand how the safety atmosphere and personal factors influence the safety behaviors of construction workers. In order to verify the relationships between discomfort in the work environment and unsafe behaviors, Chen and Li [23] added the intermediary variable of psychological emotion to verify their relationships. Based on the theory of group behavior, Jiang et al. [84] verified through SEM that the key factor in the spread of unsafe behaviors of workers is the lack of a good corporate safety culture.

**System dynamics (SD):** The second is to use SD to find the fundamental factors. Shin et al. [85] proposed an SD-based mental process model of construction workers and analyzed the feedback mechanism and the resulting dynamics on workers’ safety attitudes and safety behaviors. Kim et al. [86] studied two approaches of cognitive process based on the worker’s cognitive process model and used the SD model to investigate the possible causes of unsafe behaviors of workers. Jiang et al. [11] established a system dynamics model (SD-CUB) of causal relationships between unsafe behaviors involving management, personal, and environmental conditions to demonstrate the causes of unsafe behaviors.

**Agent-based model:** Agent-based modeling is also gradually used to explore the causes of workers’ unsafe behaviors. This method focuses more on the interaction between workers and multiple agents. Fang et al. [87] established a cognitive model of unsafe behaviors of construction workers (CM-CWUB), and conducted a systematic analysis of the cognitive failures that lead to unsafe behaviors of construction workers at different cognitive stages. Wang et al. [88] used the Cognitive Work Analysis (CWA) to describe the interaction mechanism between the work environment and personal behaviors in order to verify the relationships between discomfort in the work environment and unsafe behaviors. Li et al. [89] analyzed the complex mechanism of unsafe behaviors of construction workers and built a three-layer structure model based on the Agent modeling method and conducted a multi-agent simulation analysis. Choi et al. [44] established an agent model based on experience to investigate how workers’ social cognitive processes interact with safety management interventions and influence workers’ safety behaviors under different on-site risk conditions. Zhang et al. [90] proposed an Agent-based construction safety-related behavior modeling method, which regarded safety performance as an emergency attribute of the behaviors and interaction of construction personnel and management team. Ye et al. [91] developed an agent-based modeling method to explore the interaction of the social-cognitive process of construction workers with managers, colleagues and foremen. The authors also applied the developed model to explore the reasons for the cognitive failure of construction workers and the influence of social groups and social organization factors on workers’ unsafe behaviors [91].

In addition, there is also the hierarchical linear model (HLM), which is more complex, so the supporting literature is less, such as Wang et al. [92] established a multi-level linear model (HLM) to explore the relationship among safety atmosphere, consciousness and behavior.

Through the inductive analysis of the literature on the formation mechanism of unsafe behaviors of construction workers, it can be seen that the research method mostly used in historical studies is to construct structural equation models to verify the relationships between different factors, followed by system dynamics and agent-based construction. These studies have explored the mechanism of unsafe behaviors of construction workers from different perspectives, which helps strengthen stakeholders’ understanding of the mechanism of unsafe behaviors of workers and provides directions of suggesting safety pre-control measures.

### 3.3. Pre-Control Methods of Construction Workers’ Unsafe Behaviors

On the basis of identifying the influencing factors and investigating the formation mechanism of construction workers’ unsafe behaviors, relevant measures should be taken to deal with workers’ unsafe behaviors. Teo et al. [93] proposed a framework to promote the safe work behaviors of construction workers. The framework suggests safety improvement measures from two perspectives. The first perspective is to analyze worker dynamics from a technical perspective and on-site construction work environment, and the second one is to strengthen supervision from the perspective of organization and management and conduct management measures such as safety training [93]. De Silva and Wimalaratne [94] further integrated safety management, technology, and behavioral science in the framework of occupational safety and health. 

#### 3.3.1. Organization Internal Management Perspective

Intervention from the management perspective is undoubtedly a direct and effective measure [37]. This still needs to be discussed from the two aspects of organization and manager.

For the construction organization, we should take a series of measures to improve the safety climate, including a series of safety rules and regulations, safety education system [54]. By using the SEM model, Cavazza and Serpe [50] verified that the frequency of unsafe behaviors of construction workers who have participated in safety training is much lower than that of workers who have not. Zhou et al. [37] found that the most effective management method to improve the safety atmosphere is to formulate safety rules and conduct safety training. Tam and Fung [95] proved the impacts of mandatory safety training courses on workers’ safety attitudes and the improvement of workers’ unsafe behaviors. Darvishi et al. [96] pointed out that the adoption of a safety training observation program (STOP) is an effective way to reduce unsafe behaviors and strengthen safe work practices. Hai and Zhu [51] stated that safety education is an effective measure to eliminate safety hazards and proposed intervention measures based on the human factors engineering theory. By using safety behavior sampling techniques and Bayesian network analysis, Ghasemi et al. [97] concluded that safety training for workers is the most effective way to reduce risks. Harsini et al. [52] clarified the effects of safety education interventions on workers’ unsafe behaviors through a mixed-method study (MMR). Huang and Yang [98] concluded that the transmission and popularization of safety knowledge in construction organizations is an effective method of creating a good safety atmosphere. Choi and Li [43,44] argued from the perspective of social identity that formulating work group norms are an effective way to improve worker safety behaviors. Fargnoli and Lombardi [38] stated that when construction workers do not follow the safety rules, it is easy to cause accidents. A reasonable and effective safety training program will be helpful to improve unsafe behaviors and reduce the risk of accidents [38].

For managers, they need to take reasonable and effective management measures to reflect good leadership [62]. Lai et al. [99] found that many human resource practices are closely related to safety management results and suggested that project managers should adopt relevant human resource practices to improve the effectiveness of safety management performance of construction projects. Fang et al. [40] believed that the supervision behavior of managers can help to improve the unsafe behavior of workers. Shen et al. [45] pointed out that adopting a transformational leadership approach to encourage and support workers can improve the safety atmosphere and thereby deal with unsafe behaviors. Zaira and Hadikusumo [100] put forward that reasonable safety practices of managers can regulate workers’ unsafe behaviors. Ting et al. [101] adopted unsafe behaviors observed by front-line workers, and then adopted "Behavior-Based Safety (BBS)” to improve safety management. This management method is designed by the management team to formulate a reasonable safety-related plan.

It can be seen from the summary that from the perspective of safety management, the management is very inclined to implement safety education and training, formulate reasonable safety plans, safety systems, and strengthen supervision to enhance the internal safety atmosphere of the organization, thereby enhancing on-site organization and management capabilities. Improve the behavioral safety of workers.

#### 3.3.2. Intelligent Technology Perspective

With the continuous development of science and technology, various technologies have been applied in the construction industry to monitor the behaviors of workers. A close examination of the historical literature found that most of these studies proposed effective pre-control methods from the perspectives of construction workers’ vision and their mental state monitoring. 

**Visual angle (human skeleton model):** Han et al. [102] developed a motion capture technology based on computer vision, extracted a 3D human skeleton motion model from the video, and proposed a motion classification technology that automatically detects workers’ action. Subsequently, Han et al. [103] added a case of construction workers’ climbing ladders and collected a priori model representing unsafe actions through experiments. Additionally, this study also identified similar actions in site videos, extracted 3D human skeleton models from these videos, and combined these skeleton models [103]. The prior model is converted to the same space for motion detection. After that, Han and Lee [104] proposed a vision-based unsafe behavior detection framework, which can detect predefined unsafe actions in videos. After that, Han et al. [105] used a Kinect depth sensor to capture motion data to monitor and automatically analyze the behavior of construction workers. Additionally, it is also proposed that the choice of the human skeleton model has a significant impact on motion classification and detection [105]. On this basis, Han et al. [106] proposed a modeling and classification method for identifying unsafe behaviors. By studying three types of motion data and estimating the average trajectory of the motion, it automatically recognizes the actions of workers [106]. Yu et al. [107] proposed a method based on image skeleton parameterization to identify unsafe behaviors of construction workers in real-time and conducted experiments involving three unsafe behaviors to test its feasibility and determined the range of relevant key parameters. Ji et al. [108] developed a two-dimensional human skeleton parameterization method to understand the behaviors of construction workers, and also build a behavior detection database based on image skeletons to identify the behaviors of workers.

**Visual angle (location and trajectory tracking):** Guo et al. [109] combined building information modeling (BIM) and radio frequency identification technology (RFID) to propose an early warning system for unsafe behaviors of construction workers on-site. Li et al. [110] introduced the development and application of a real-time location system (RTLS) based on chirp spread spectrum (CSS) technology to track the real-time location of construction workers on construction sites. The motion trajectory prediction model developed by Rashid and Behzadan [111] can reliably detect unsafe motion and recent collision events. On this basis, Rashid and Behzadan et al. [112] studied two trajectory prediction models, namely Polynomial regression (PR) and the hidden Markov model (HMM). Through these two models, unsafe motion and impending collisions can be reliably tested. In order to understand the worker movement in a dynamic construction environment, Arslan et al. [113] proposed a Worker Trajectory Analysis System (WoTAS). Chen and Luo [114] proposed a positioning system accuracy model and three safety clearance models to track and evaluate the behaviors of workers and site operating conditions. Arslan et al. [115] developed a data model-based intrusion detection system that can use space technology to track a worker’s position changes in the building space in real-time. Jeelani et al. [116,117] proposed a vision-based system that uses workers’ first perspective (FPV) to estimate their locations on construction sites and identify subsequent hazards.

**Visual Angle (Convolutional Neural Network):** Patel and Jha [118] used artificial neural networks to build models, identified key factors that lead to unsafe behaviors, and suggested corresponding improvement strategies to deal with these behaviors. Liu et al. [119] applied Convolutional Neural Network (CNN) to human body detection and pose estimation in sequence images under field conditions. This study designed a method for human pose estimation in a dynamic and cluttered environment [119]. In the follow-up research, Guo et al. [120] simplified dynamic motion to static posture to identify unsafe behaviors. Fang et al. [65] developed a computer vision-based automation method that uses two convolutional neural network (CNN) models to determine whether workers wear protective equipment when performing work at high places. Ding et al. [121] developed a new hybrid deep learning model by combining convolutional neural network (CNN) and long-term short-term memory (LSTM) to automatically recognize the unsafe behaviors of workers. Fang et al. [122] used a masked region-based convolutional neural network (R-CNN) to detect the relationships between individual workers and the building structure in subsequent research, which helps automatically identify unsafe behaviors.

**Visual angle (image big data):** Guo et al. [123,124] introduced behavior observation based on big data to accurately identify unsafe behaviors of on-site workers. The work breakdown structure (WBS) forms a behavior risk knowledge base, and Work Hazard Analysis (JHA) is adopted to analyze influencing factors [123,124]. Finally, Vector Space Model (VSM) is used to match and classify influencing factors with pre-defined unsafe behaviors so that construction organizations can visualize unsafe behaviors and make judgments in real-time [123,124]. Liu et al. [125] introduced an application based on intelligent behavior recognition technology, which can detect the behaviors of workers more accurately through clearer image data.

**Visual angle (virtual reality technology):** Chun et al. [126] used the construction virtual prototype (CVP) to create a virtual environment in which construction workers can explore and identify construction hazards. At the same time, in a dangerous situation, the simulation of a worker’s posture has also been greatly changed compared with the traditional on-site monitoring method. Shi et al. [69,127] used a multi-user virtual reality (MVR) system with a motion tracking function to simulate dangerous scenes and studied the performance of construction workers in dangerous scenes such as seeing other workers working at heights. The study also investigated whether a worker’s own behavioral safety will be affected by these dangerous scenes [69,127].

**Psychological monitoring perspective (smart wearable device):** Hirokane and Kamijo [128] used a wristwatch-style pulse and heart rate measurement device to monitor workers’ unsafe behaviors caused by mental status. Guo et al. [129] proposed an efficient wearable technology-based method to collect workers’ psychological data and studied unsafe behaviors through their psychological status. 

**Psychological monitoring angle (EEG signal-EEG):** Chen et al. [130] designed an on-site experiment and proposed a quantitative detection method, which monitors workers’ activity by processing real-time recorded EEG signals and decomposing them through wavelet packets. On this basis, Wang et al. [131] proposed a new hybrid kinematics-EEG data type, and used EEG wavelet packet decomposition to calculate alert measurement indicators and identify appropriate signal subbands used to detect the alert level of construction workers.

**Combination of visual angle and psychological monitoring angle:** Migliaccio et al. [132] used the data fusion method to continuously and remotely monitor the location and health of construction workers. Additionally, Cheng et al. [133] adopted non-intrusive real-time worker location perception (RTLS) and psychological state monitoring (PSM) technology to analyze data to improve unsafe behaviors in time. Yu et al. [107] studied the relationships between workers’ mental status and unsafe behaviors based on the virtual reality technology.

It can be seen that vision-based intelligent technology is widely used in the field of construction, especially the use of vision technology to predict the movements and postures of workers to prevent unsafe behaviors. In addition, smart devices for measuring and controlling the mental status of workers have gradually become a trend in current research. With the advancement of science and technology and the promotion of smart technology, the behaviors of workers can be better monitored, and the incidence of unsafe behavior is gradually decreasing.

## 4. Discussion

On the basis of the state-of-the-art review, future research directions in the “unsafe behaviors of construction workers” field were suggested.

When exploring influencing factors of construction workers’ unsafe behaviors, most of the historical studies focus on these factors at the individual level, such as investigating individual behaviors, their behavior correction, and improvement of their psychological characteristics. Additionally, most studies that investigated unsafe behaviors from the perspective of organizational management also focus on the improvement of individual behaviors. It should be noted that construction workers’ different characteristics (e.g., ages, educational background, and cognitive abilities) impacts their unsafe behaviors. Research findings of unsafe behaviors based on individual characteristics (e.g., attitudes, awareness, risk perception capabilities) cannot reflect that of the group of construction workers as a whole. Therefore, exploration of the impacts of group characteristics of construction workers on unsafe behaviors is warranted. From the perspective of "organizational management factors", managers can use the "Sensitive leadership" approach, in-depth exchanges and interviews with workers within the worker group to have a deeper understanding of this issue.

Regarding the formation mechanism of unsafe behaviors, current research focuses on constructing models for analysis from one or more aspects of individual worker factors, organizational management factors, and production operation factors. Given that the entire formation process of unsafe behaviors of construction workers is complicated, it is difficult to explain the complete evolution process only by structural equation model (SEM) and system dynamics (SD). Moreover, due to the differences of workers, the "multi-agent" involved increases its complexity. Therefore, when combining some behavioral theories in the future investigations, the method of "multi-agent modeling" can be used to study complex group behaviors, and continuously observe output changes through sensitivity analysis to verify the rationality of the assumptions in the formation mechanism. However, due to the hierarchical nature of management, some impacts across levels cannot be fed back in time [38]. Therefore, the multi-level linear models (HLM) are suggested to be used to further explore the complexity between levels in future research. 

In terms of the pre-control methods, managers are most inclined to provide safety education and training to workers [11], and safety education and training is also the most effective safety intervention method [50]. Most of the current safety education and training are compulsory and are not something workers are willing to learn about. Therefore, the future safety management research can explore the use of some methods that are better than traditional methods (e.g., computer-aided technology). Improve the subjective initiative of workers to improve the effectiveness of construction safety training [134], and then conduct a comparative study to further verify its effectiveness through some data and methods (such as factor analysis, etc.). Additionally, as most of the current research proposes related measures from a visual perspective and explorations from the perspective of workers’ mental status is still lacking [131], the future research is suggested to explore the use of technologies and equipment to address this research gap. 

## 5. Conclusions

The unsafe behaviors of construction workers have attracted considerable attention in the construction management research community. This study conducted a systematic literature review about research on “unsafe behaviors of construction workers” by using 140 academic papers. A content analysis of these papers found that the main topics include the influencing factors, formation mechanism, and pre-control measures of workers’ unsafe behaviors. More specifically, the identified influencing factors of workers’ unsafe behaviors can be divided into three groups, including individual factors, organizational management factors, and production operation factors. Regarding the formation mechanism of workers’ unsafe behaviors, scholars use different methods to analyze the interaction between various factors and explore the causes of unsafe behavior. In terms of pre-control methods, most studies suggested measures from the perspectives of management and intelligent technology to as early as possible to curb the occurrence of unsafe behavior. Although the current research on unsafe behaviors of construction workers has achieved fruitful results, research gaps can still be found. Future research directions have also been suggested in this study, it includes the following three aspects:The construction workers are a huge group, and there are many differences among the workers. At present, the research focuses on the individual influencing factors of the workers, and ignores the influence of the group characteristics of the workers on the occurrence of unsafe behaviors to a certain extent. Therefore, when studying the influencing factors of the unsafe behaviors of the construction workers in the future, the industry stakeholders can consider paying more attention to the group characteristics of the workers.The formation of unsafe behavior of construction workers is a complex dynamic process with multi variables, multi-dimensions and interaction. At present, most of the research on the formation mechanism adopts SEM or SD, usually starting from a single subject such as individual workers or organizational management, and the complete evolution process involves multiple levels of individual, organization, and environment. In future research, “Multi-Agent Modeling” and “Multi-Layer Linear Model” can be used to better explore the relationship between multiple agents and different levels.In the research of preventive measures for workers’ unsafe behavior, vision-based technology has achieved great success. At the same time, the research of workers’ psychological monitoring equipment should be better discussed in the future.

The research provides valuable implications for construction stakeholders to improve their safety performance in practice. It also guides the future research in the studied field. 

## Figures and Tables

**Figure 1 ijerph-18-02644-f001:**
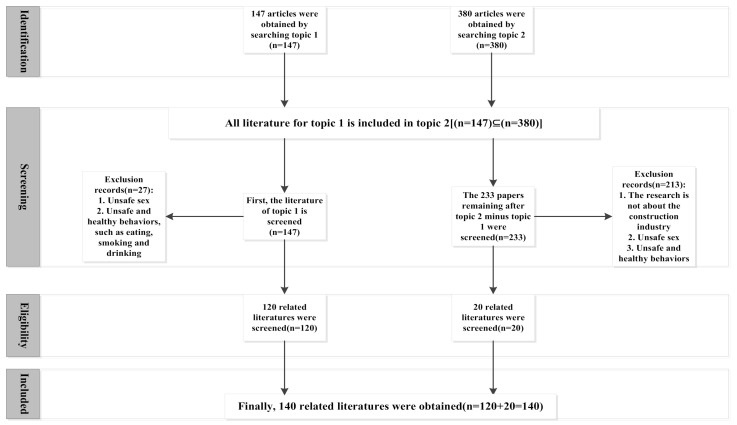
Flowchart of the systematic review process (PRISMA flow diagram).

**Figure 2 ijerph-18-02644-f002:**
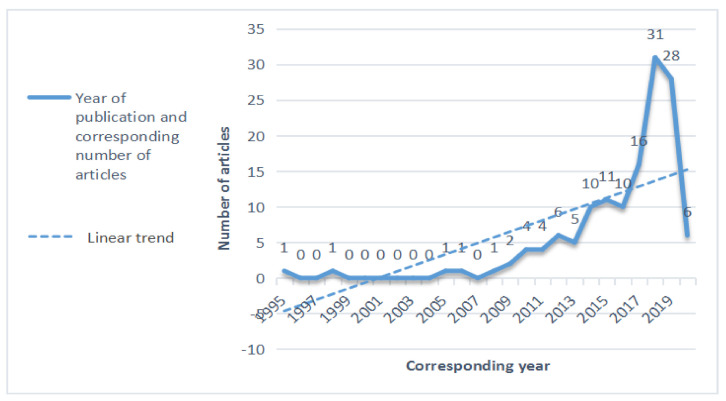
The number of publications in different years (1995–2020).

**Figure 3 ijerph-18-02644-f003:**
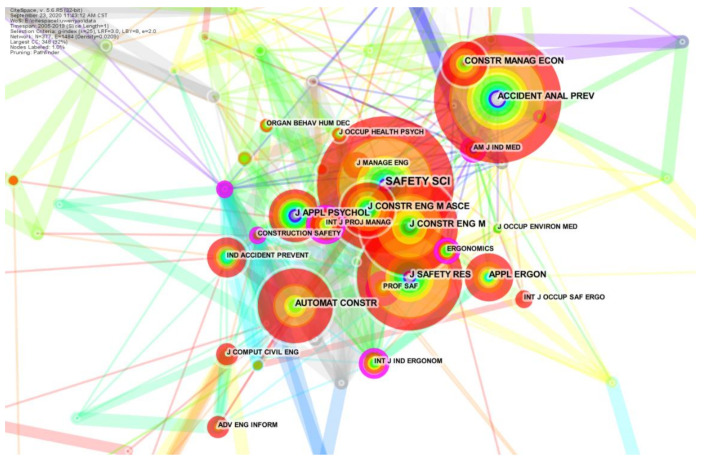
The result of literature publication source analysis.

**Figure 4 ijerph-18-02644-f004:**
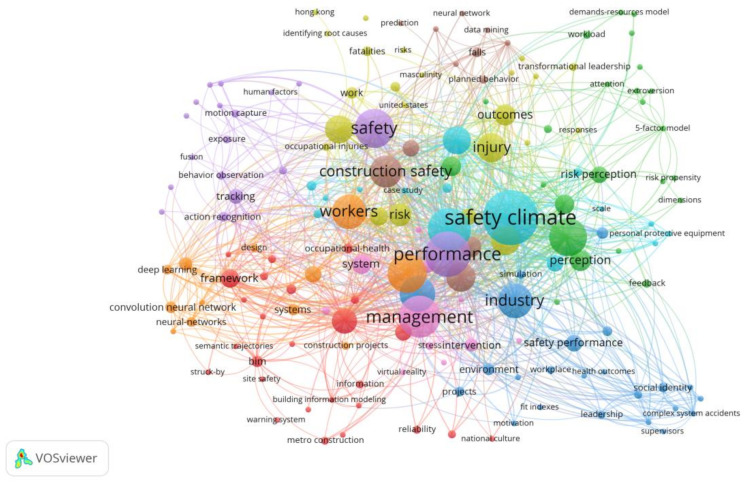
The cluster graph of keyword co word analysis.

**Figure 5 ijerph-18-02644-f005:**
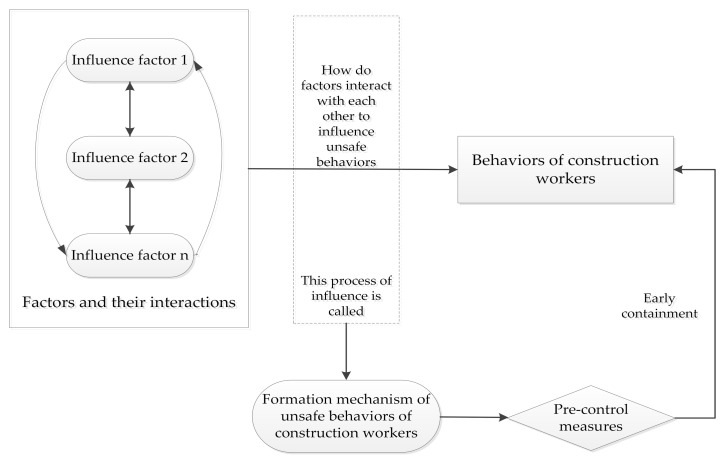
The research framework in the studied field.

**Figure 6 ijerph-18-02644-f006:**
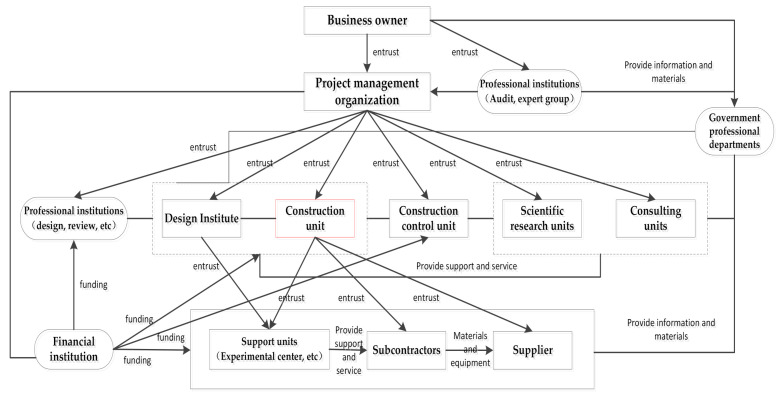
Interaction between construction subject and its stakeholders.

**Table 1 ijerph-18-02644-t001:** The clustering results of keywords co word analysis.

Cluster 1	Cluster 2	Cluster 3
Safety Climate; Risk perception; Risk propensity; Feed back; Responses; Attention; Dimensions; Workload; Demands-resources model; 5-factor model; Scale; Simulation; Case study	Performance; Safety; (Construction Safety); Tracking; Action recognition; Behavior recognition; Fusion; Exposure; Motion capture; Human factors; Neural network ; Data mining; Prediction; Falls; Planned behavior	Management; System; Framework; Bim; Occupational-health; Site safety ; Information; Building information modeling; Warning System; Metro construction; Virtual reality; Reliability
**Cluster 4**	**Cluster 5**	**Cluster 6**
Industry; Safety performance; Environment; Workplace; Health; Projects; Social identity; Fit indexes; Leadship; Complex system accidents; Motiration; Supervisors; Personal protective; equipment	Workers; Construction project; Design; Deep learning; Neural-networks; Semantic Trajectories; Struck-by	Injury; Risk; Occupational Injury; Masculinity; Transformational-leadership; Fatalities; Identifying root Causes; Work; Hong-kong

**Table 2 ijerph-18-02644-t002:** Individual factors.

Major Categories	Category Segmentation	Source
Individual Factors	Physiological Factors	Yang and Byung [16], Fang et al. [17], Syamlal et al. [18]
	Psychological Factors	Yang and Byung [16], Leung et al. [19], Kim [20], Ju et al. [21], Wang et al. [22], Chen and Li [23], Wu et al. [24], Mohammadi and Tavakolan [2]
	Personality characteristics	Sing et al. [25], Chen et al. [26], Hasanzadeh et al. [27], Hasanzadeh et al. [3], Zhang et al. [4]
	Subjective attitude consciousness	Cavazza and Serpe [28], Xu et al. [29], Gyu-sun et al. [30]
	Risk perception	Burns and Conchie [31], Huang et al. [32], Man et al. [33]
	Language and cultural barriers	Al-Bayati et al. [34], Al-Bayati et al. [35], Chan et al. [36]

**Table 3 ijerph-18-02644-t003:** Organizational management factors.

Major Categories	Category Segmentation		Source
Organizational management factors	Safety Climate	Management’s obligations	Zhou et al. [37], Fargnoli and Lombardi [38], He et al. [39]
		Worker safety participation	He et al. [39], Fang et al. [40]
	Group norms		Arcury et al. [41], Choi and Lee [42], Choi and Lee [43], Choi et al. [44]
	Leadership		Fang et al. [40], Shen et al. [45], Xiong et al. [46]
	Management methods		Du et al. [47], Sheng et al. [48], Choudhry [49], Li et al. [9], Cavazza and Serpe [50], Hai and Zhu [51], Harsini et al. [52]

**Table 4 ijerph-18-02644-t004:** Project level factors.

Major Categories	Category Segmentation	Source
Project level factors	Safety investment	Kim and Park [56], Fang et al. [40]
	Safety inspection and feedback	Fernández-Muñiz et al. [57], Tam et al. [58], Teo and Ling [59], Nielsen [60], Iyer et al. [61], Mohamed [62]

**Table 5 ijerph-18-02644-t005:** Production and operation factors.

Major Categories	Category Segmentation	Source
Production and operation factors	Operation mode	Johnson et al. [63], Kaskutas et al. [64], Fang et al. [65], Kolar et al. [66], Yin et al. [67], Shokouhi et al. [68], Shi et al. [69], Niu and Chen [70], Eskisar et al. [71]
	Working environment	Chi et al. [72], Jiang et al. [11], Lu and Davis [73], Chen et al. [74], Fang et al. [40], Mohamed et al. [75], Del Puerto et al. [76]
	Construction equipment	Zhao et al. [77], Niu and Chen [70], Kaskutas et al. [78], Li et al. [79], Zhang et al. [80]

## Data Availability

The data presented in this study are available on request from the corresponding author.

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
