# Peer review of "Influencing Factors, Mechanism and Prevention of Construction Workers’ Unsafe Behaviors: A Systematic Literature Review"

_ijerph, 2021, doi:10.3390/ijerph18052644_

Round 1

Reviewer 1 Report

The authors conducted a very interesting systematic review conducted with the right methodology. I suggest that future developments to improve the conditions of these workers be included in the conclusions. 

Author Response

Response to Reviewer 1 Comments

Point1:

The authors conducted a very interesting systematic review conducted with the right methodology. I suggest that future developments to improve the conditions of these workers be included in the conclusions. 

Response1:

Thank you for your valuable suggestions, which help us to further improve the integrity of the conclusion. The revised conclusions are as follows:

The unsafe behaviors of construction workers have attracted considerable attention in the construction management research community. This study conducted a systematic literature review about research on "unsafe behaviors of construction workers" by using 140 academic papers. A content analysis of these papers found that the main topics include the influencing factors, formation mechanism, and pre-control measures of workers’ unsafe behaviors. More specifically, the identified influencing factors of workers’ unsafe behaviors can be divided into three groups, including individual factors, organizational management factors, and production operation factors. Regarding the formation mechanism of workers’ unsafe behaviors, scholars use different methods to analyze the interaction between various factors and explore the causes of unsafe behavior. In terms of pre-control methods, most studies suggested measures from the perspectives of management and intelligent technology to as early as possible to curb the occurrence of unsafe behavior. Although the current research on unsafe behaviors of construction workers has achieved fruitful results, research gaps can still be found. Future research directions have also been suggested in this study,it includes the following three aspects:

â‘ The construction workers are a huge group, and there are many differences among the workers. At present, the research focuses on the individual influencing factors of the workers, and ignores the influence of the group characteristics of the workers on the occurrence of unsafe behaviors to a certain extent. Therefore, when studying the influencing factors of the unsafe behaviors of the construction workers in the future, the industry stakeholders can consider paying more attention to the group characteristics of the workers.

②The formation of unsafe behavior of construction workers is a complex dynamic process with multi variables, multi dimensions and interaction. At present, the research on the formation mechanism mostly adopts SEM or SD, which is often from a single subject of individual workers or organizational management. The complete evolution process involves multiple levels such as individual and organizational management In the future research, “Multi-Agent Modeling” and “Multi-Layer Linear Model” can be used to better explore the relationship between multiple agents and different levels.

③In the research of preventive measures for workers' unsafe behavior, vision based technology has achieved great success. At the same time, the research of workers' psychological monitoring equipment should be better discussed in the future.

The research provides valuable implications for construction stakeholders to improve their safety performance in practice. It also guides the future research in the studied field.

Yours sincerely,

Zhen Li

Name:Zhen Li

E-mail:janeli@ujs.edu.cn

Cc.

Qingfeng Meng,Wenyao Liu,Xin Hu

Reviewer 2 Report

The authors of this manuscript conducted a systematic literature review about influencing factors, mechanism and prevention of construction workers’ unsafe behaviors. In general, it is a well-written manuscript. The literature review is thorough. The flowchart (or the way that the authors classified their review process) seems quite logical. To improve the manuscript, I would like to make some suggestions.

  1. I would like to suggest the authors to have a project level when studying the factors affecting construction workers’ unsafe behaviors (3.1). As the authors know, even for a same organization (say a general contractor), employees’ safety behavior depend on other stakeholders. Indeed, safety climate or safety circumstances (or norms in a construction site) is heavily influenced by project owners (from the perspective of general contractors) or general contractors (from the perspective of subcontractors). Organizational level is about an intra-organizational matter but project level is more about an inter-organizational matter as there are many stakeholders involved in a single project. (I also think that the project-level should be between organizational level and production and operational factor level.)
  2. For some sections such as 3.2. and 3.3.1, the authors just listed the previous studies so it is hard to read. Providing a table (for each section) for summary may help readers better understand the paper.
  3. Two typos: line 81 (2010? As Figure 2 includes 2019, I thought it is a typo), line 164 ([References?]

Author Response

Response to Reviewer 2 Comments

Thank you for your valuable advice, your suggestion helps us to better improve the manuscript, so that the content of the manuscript has been improved, we have revised and improved one by one according to your valuable advice.

Point1:

I would like to suggest the authors to have a project level when studying the factors affecting construction workers’ unsafe behaviors (3.1). As the authors know, even for a same organization (say a general contractor), employees’ safety behavior depend on other stakeholders. Indeed, safety climate or safety circumstances (or norms in a construction site) is heavily influenced by project owners (from the perspective of general contractors) or general contractors (from the perspective of subcontractors). Organizational level is about an intra-organizational matter but project level is more about an inter-organizational matter as there are many stakeholders involved in a single project. (I also think that the project-level should be between organizational level and production and operational factor level.)

Response1:

Thank you for your valuable opinions. These opinions help us to make a more detailed division when analyzing the influencing factors of unsafe behaviors. According to your opinions, we add a project level influencing factor and draw a diagram to demonstrate our narration. Specific figures, tables and descriptions are as follows:

As we all know, the completion of an engineering project requires the cooperation of multiple organizations. In the construction process, there is an interactive relationship between the construction subject (construction unit) and many stakeholders (as shown in Figure 6). The whole process has a clear hierarchy. Only through the entrustment of the upstream organization and the orderly cooperation of the downstream organization, can the project be successfully completed Organization interaction, which also makes the safety behavior of workers in the construction subject depend on other organizations to a certain extent. Therefore, from the perspective of the project as a whole, the safety problems between organizations should also be discussed, and the safety problems between organizations need to start from the level and interaction between organizations. Table 4 lists two influencing factors at the project level, namely safety input and safety inspection feedback.

Figure 6.Interaction between construction subject and its stakeholders.

As the picture can't be displayed here, you can open the attachment to read the detailed picture.

Table 4.Project level factors

Major Categories

Category Segmentation

Source

Project level factors

Safety investment

Kim and Park [129],Fang et al. [38]

Safety inspection and feedback

Fernández-Muñiz et al.[130],Tam et al. [131],

Teo and Ling [132],Nielsen[133],Iyer et al. [134],Mohamed [135]

Safety investment is the most basic guarantee for safety production. Due to the hierarchical safety of construction projects, insufficient funds and large arrears of safety investment, the coordinated operation of each level will be affected. Kim and Park [129] believes that the inadequate investment of workers' safety equipment and safety facilities in the workplace is the main cause of safety accidents. Due to the lack of safety investment, the shortage of funds for safety education and training of workers, the level of safety knowledge can not be improved, which also increases the incidence of safety accidents [38].

Due to the hierarchical nature of engineering project organizations, downstream organizations are often subject to the supervision of upstream organizations. The higher-level project management agency will regularly send personnel to the lower-level construction units to conduct safety inspections. In this process, the upper-level management agency will further popularize safety for the lower-level construction units Policies and standards [130,131], analysis and prevention of existing safety hazards [132-133], while the upper management organization will also listen to the feedback information of the construction unit [134], so as to conduct a comprehensive safety management system audit [135]. The interaction of managers between different organizations also contributes to the exchange of management experience [134], thereby better promoting the order and safety of the construction site. This kind of communication and interaction between organizations starting from the project as a whole has greatly improved the safety performance of the entire project.

Point2:

For some sections such as 3.2. and 3.3.1, the authors just listed the previous studies so it is hard to read. Providing a table (for each section) for summary may help readers better understand the paper.

Response2:

Thank you for your valuable comments. We have made the following improvements.

For 3.2, we have made the following improvements:

In order to better find the key causes of unsafe behavior, scholars take the workers as the core subject, and constantly combine a series of influencing factors around the workers to explore, find the factors with greater influence effect and make improvement. As the research content is diverse and involves many factors at different levels, the occurrence mechanism is divided and analyzed from the perspective of the research method.

Structural equation modeling (SEM): In the field of safety, it is the most common to use SEM to analyze the relationship between variables (factors).Khosravi et al. [66] constructed a SEM model from the perspective of safety supervisors for workers’ individual factors, safety atmosphere and environmental conditions in the workplace, and concluded that the safety status of the workplace plays crucial roles in linking the safety atmosphere and worker participation. Goh and Sa'adon [69] used the Planned Behavior Theory (TPB) to model the cognitive factors that affect the behavior of workers at heights, and studied them and determined which TPB structure (attitude , subjective norms, perceived behavior control and intention) is potentially important influencing factors of safety behaviors at heights. In order to determine the key factors affecting workers’ safety risk tolerance.Guo et al. [71] developed and tested eight competition models related to the safety behaviors of construction workers to better understand how the safety atmosphere and personal factors influence the safety behaviors of construction workers. In order to verify the relationships between discomfort in the work environment and unsafe behaviors, Chen and Li [17] added the intermediary variable of psychological emotion to verify their relationships. Jiang et al. [75] Based on the group behavior theory, SEM is constructed to verify that the most critical communication of unsafe behavior of construction workers is due to the lack of safety culture.

System dynamics (SD): The second is to use SD to find the fundamental factors.Shin et al. [67] proposed a SD-based mental process model of construction workers and analyzed the feedback mechanism and the resulting dynamics on workers’ safety attitudes and safety behaviors. Lee et al. [68] studied two approaches of cognitive process based on the worker's cognitive process model and used the SD model to investigate the possible causes of unsafe behaviors of workers.Jiang et al. [6] established a system dynamics model (SD-CUB) of causal relationships between unsafe behaviors involving management, personal, and environmental conditions to demonstrate the causes of unsafe behaviors.

Agent based model: Agent based modeling is also gradually used to explore the causes of workers' unsafe behaviors. This method focuses more on the interaction between workers and multiple agents.Fang et al. [70] established a cognitive model of unsafe behaviors of construction workers (CM-CWUB), and conducted a systematic analysis of the cognitive failures that lead to unsafe behaviors of construction workers at different cognitive stages.Wang et al. [72] used the Cognitive Work Analysis (CWA) to describe the interaction mechanism between the work environment and personal behaviors. In order to verify the relationships between discomfort in the work environment and unsafe behaviors.Li et al. [74] analyzed the complex mechanism of unsafe behaviors of construction workers and built a three-layer structure model based on the Agent modeling method and conducted a multi-agent simulation analysis.Choi et al. [37] established an agent model based on experience to investigate how workers' social cognitive processes interact with safety management interventions and influence workers' safety behaviors under different on-site risk conditions.Zhang et al. [76] proposed an Agent-based construction safety-related behavior modeling method, which regarded safety performance as an emergency attribute of the behaviors and interaction of construction personnel and management team.Ye et al. [77] developed an agent-based modeling method to explore the interaction of the social cognitive process of construction workers with managers, colleagues and foremen. The authors also applied the developed model to explore the reasons for the cognitive failure of construction workers and the influence of social groups and social organization factors on workers' unsafe behaviors [77].

In addition, there is also the hierarchical linear model (HLM), which is more complex, so the supporting literature is less, such as Wang et al. [73] established a multi-level linear model (HLM) to explore the relationship among safety atmosphere, consciousness and behavior.

For 3.3.1, we have made the following improvements:

Intervention from the management perspective is undoubtedly a direct and effective measure [31].This still needs to be discussed from the two aspects of organization and manager.

For the construction organization, we should take a series of measures to improve the safety climate, including a series of safety rules and regulations, safety education system[124].By using the SEM model, Nicoletta and Alessandra [45] verified that the frequency of unsafe behaviors of construction workers who have participated in safety training is much lower than that of workers who have not.Zhou et al. [31] found that the most effective management method to improve the safety atmosphere is to formulate safety rules and conduct safety training.Vivian and Ivan [80] proved the impacts of mandatory safety training courses on workers' safety attitudes and the improvement of workers' unsafe behaviors.Darvishi et al. [81] pointed out that the adoption of a safety training observation program (STOP) is an effective way to reduce unsafe behaviors and strengthen safe work practices.Hai and Zhu [46] stated that safety education is an effective measure to eliminate safety hazards and proposed intervention measures based on the human factors engineering theory.By using safety behavior sampling techniques and Bayesian network analysis, Fakhradin et al. [82] concluded that safety training for workers is the most effective way to reduce risks.Azita et al. [47] clarified the effects of safety education interventions on workers' unsafe behaviors through a mixed method study (MMR).Huang and Yang [83] concluded that the transmission and popularization of safety knowledge in construction organizations is an effective method of creating a good safety atmosphere.Choi and Li [36~37] argued from the perspective of social identity that formulating work group norms are an effective way to improve worker safety behaviors. Mario and Mara [32] stated that when construction workers do not follow the safety rules, it is easy to cause accidents. A reasonable and effective safety training program will be helpful to improve unsafe behaviors and reduce risk of accidents [32].

For managers, they need to take reasonable and effective management measures to reflect good leadership[135].Diana et al. [84] found that many human resource practices are closely related to safety management results and suggested that project managers should adopt relevant human resource practices to improve the effectiveness of safety management performance of construction projects.Fang et al. [38] believed that the supervision behavior of managers can help to improve the unsafe behavior of workers.Shen et al. [39] pointed out that adopting a transformational leadership approach to encourage and support workers can improve the safety atmosphere and thereby deal with unsafe behaviors.Mohamed and Bonaventura [85] put forward that reasonable safety practices of managers can regulate workers' unsafe behaviors.Ting et al. [86] adopted unsafe behaviors observed by front-line workers, and then adopted "Behavior-Based Safety (BBS) ” to improve safety management. This management method is designed by the management team to formulate a reasonable safety-related plan.

Point3:

Two typos: line 81 (2010? As Figure 2 includes 2019, I thought it is a typo), line 164 ([References?]

Response3:

Line81:

Sorry, this is a small mistake in our work. We have changed the retrieval year to 1986-2020.

Line 164:

I'm very sorry, this is a small mistake in our work. The reference of this passage is from Yang and Kim [12].

[12].Yang, Yong Koo.,Byung-Seok, Kim.Study on the Structural Relation between the Level of Fatigue and Stress of Construction Workers and Disaster Risks.J. Korea .Saf. Manag & Sci.2014,16,35-44.doi:10.12812/ksms.2014.16.3.35.

Once again, we would like to express our sincere thanks for your review and suggestions on the manuscript. Your suggestions have greatly helped us and we have learned a lot about safety. Thank you for your suggestions.

Yours sincerely,

Zhen Li

Name:Zhen Li

E-mail:janeli@ujs.edu.cn

Cc.

Qingfeng Meng,Wenyao Liu,Xin Hu

Reviewer 3 Report

This reviewer believes that the authors did a fair job in the study. However, a significant revision is needed to make the paper stronger.  The authors should carefully address the following concerns: 

  • References 1 and 2 are outdated, while the whole article has been built on them. Recent references must be used to support the author's claim. 
  • Line 81: the literature searching years should be changed to 186-2020. 
  • Line 153: you stated that safety climate factors as one level of the factors without providing a clear definition for the meaning of the safety factor. Furthermore, you have used the safety culture terminology for the whole article instead of the safety climate. This is clear interchangeable use of these two terminologies, which has been discussed in a recent publication by Al-Bayati et al. (2019). It is recommended to discuss this issue carefully in your work and provide a clear definition for both terminologies.

Al-Bayati, A. J., Albert, A. and Ford, G. (2019) “Construction Safety Culture and Climate: Satisfying the Necessity for an Industry Framework." Practice Periodical on Structural Design and Construction, 24 (4), DOI:10.1061/(ASCE)SC.1943-5576.0000452

  • The authors discussed individual factors on page 7, but they missed a critical factor: language and culture barriers. See the following references: 
  •  human factors what about language and cultural
    barriers? check the following: 
    • Al-Bayati, A. J., Abudayyeh, O., Fredericks, T., and Butt, S. (2017) "Managing Cultural Diversity at U.S. Construction Sites: Hispanic Workers' Perspectives." Journal of Construction Engineering and Management, DOI:10.1061/(ASCE)CO.1943-7862.0001359
    • Chan, A., Javed, A., Lyu, S., Hon, C., and Wong, F. (2016). "Strategies for improving safety and health of ethnic minority construction workers." J. Constr. Eng. Manage.,10.1061/(ASCE)CO.1943-7862.0001148,05016007.
  • The authors should also state in the introduction section as well as in the discussion section that "A human action that precursors a work-related accident could be a system error - not a human error." When organizational management factors (line 225) were the root cause (Al-Bayati et al. 2018). 
    • Al-Bayati, J., Abudayyeh, and Albert, A. (2018). “Managing Active Cultural Differences in U.S. Construction Workplaces: Perspectives from Non-Hispanic Workers." Journal of Safety Research, Elsevier. DOI: 10.1016/j.jsr.2018.05.004
  • Line 164. There is a reference missing. 
  • The authors should clearly explain the differences between fatigue and pressure. For example, culture and language barriers lead to pressure; could they lead to fatigue as well? 

I look forward to reviewing the revised version of the article. 

Good Luck!

Author Response

Response to Reviewer 3 Comments

Thank you for your valuable advice, your suggestion helps us to better improve the manuscript, so that the content of the manuscript has been improved, we have revised and improved one by one according to your valuable advice.

Point1:

References 1 and 2 are outdated, while the whole article has been built on them. Recent references must be used to support the author's claim. 

Response1:

In order to make references more timely, we add references [19], [23], [24], [121], [122] to provide more timely support. They all summarize the current situation of the construction industry at the beginning of the article, and mention that the development of the construction industry has been plagued by frequent accidents and injuries, and most of the causes of accidents come from people's unsafe behaviors. The sources of literature are as follows:

[19].Mohammadi, A.,Tavakolan, M.Modeling the effects of production pressure on safety performance in construction projects using system dynamics.J. Saf. Res.2019,71,273-284,doi:10.1016/j.jsr.2019.10.004.

[23].Hasanzadeh, Sogand.,Dao, Bac.,Esmaeili, Behzad.,Dodd, Michael D.Role of Personality in Construction Safety: Investigating the Relationships between Personality, Attentional Failure, and Hazard Identification under Fall-Hazard Conditions.J. Constr. Eng. Manag.2019,145,10.1061/(asce)co.1943-7862.0001673.

[24].Zhang, J.,Xiang, P. C.,Zhang, R.,Chen, D.,Ren, Y. T.Mediating Effect of Risk Propensity between Personality Traits and Unsafe Behavioral Intention of Construction Workers.J. Constr. Eng. Manag.2020,146,doi;10.1061/(asce)co.1943-7862.0001792.

[121].Feldmann, D., and K. H. Welge. Understanding the causation of construction workers’ unsafe behaviors based on system dynamics modeling. J. Manage. Eng. 2014,31 (6): 04014099,doi:10.1061 /(ASCE)ME.1943-5479.0000350.

[122].Shuquan Li ; Xiuyu Wu, S.M.ASCE ; Xuezhao Wang ; and Songhe Hu.Relationship between Social Capital, Safety Competency, and Safety Behaviors of Construction Workers.J. Constr. Eng. Manag.2020,146(6),doi:10.1061/(ASCE)CO.1943-7862.0001838.

Point2:

Line 81: the literature searching years should be changed to 1986-2020. 

Response2:

Sorry, this is a small mistake in our work. We have changed the retrieval year to 1986-2020.

Point3:

Line 153: you stated that safety climate factors as one level of the factors without providing a clear definition for the meaning of the safety factor. Furthermore, you have used the safety culture terminology for the whole article instead of the safety climate. This is clear interchangeable use of these two terminologies, which has been discussed in a recent publication by Al-Bayati et al. (2019). It is recommended to discuss this issue carefully in your work and provide a clear definition for both terminologies.

Al-Bayati, A. J., Albert, A. and Ford, G. (2019) “Construction Safety Culture and Climate: Satisfying the Necessity for an Industry Framework." Practice Periodical on Structural Design and Construction, 24 (4), DOI:10.1061/(ASCE)SC.1943-5576.0000452

Response3:

Thank you very much for your suggestion, which helps us to further explore the relationship between safety culture and safety climate. In the reference you provided, Al Bayati et al. Proposed the difference between safety culture and safety climate. In the second paragraph of the literature review, the author mentioned that Flin et al. Believed that "safety climate is the expression of value norms and beliefs including basic safety culture". The author also believed that safety culture will affect the safety climate level in the workplace. After discussion, we think that The safety culture should be included in the safety climate, which is an embodiment of the safety climate level.

Through further collation of the literature, we found that N. Dedobbleer et al.[124] Proposed that "the two major factors of safety climate should be the management's obligations (e.g. safety commitment, safety compliance, safety training) and workers' safety participation"; and the definition of safety culture in 2008 by NCA is "how safety culture is seen to operate with safety and health and relative to other workplaces" The definition of "basic organizational principles, norms, commitments and values related to the relative importance of objectives" is consistent with the "obligations of management" proposed by N. Dedobbleer[124] and others, and also confirms the inclusion of safety climate for safety culture. Therefore, we have changed the table and narration in the original manuscript. The four influencing factors of organizational management are divided into Safety climate, Group norms, Leadership and Management methods. The influencing factor of Safety climate is divided into two dimensions: Management's observations and Worker safety participation, and a new narration is made. The corresponding table and narration are as follows:

Table 3. Organizational management factors.

Major Categories

Category Segmentation

Source

Organizational management factors

Safety Climate

Management's obligations

Zhou et al. [31],Mario and Mara [32],He et al. [33]

Worker safety participation

He et al. [33],Fang et al. [38]

Group norms

Thomas et al. [34],Choi and Lee [35],Choi and Lee [36],Choi et al. [37]

Leadership

Fang et al. [38],Shen et al. [39],Xiong et al. [40]

Management methods

Du et al. [41],Sheng et al. [42],Choudhry[43],Li et al. [44],Nicoletta and Alessandra [45],Hai and Zhu [46],Azita et al. [47]

  1. Zohar [123] believes that safety climate is the concern of employees on safety, and this concern is the perception of integrity. N. Dedobbleer et al. [124] proposed that the two major factors of safety climate in construction industry should be the obligation of management (e.g. safety commitment, safety compliance, safety training) and workers' safety participation. In 2008, NCA defined safety culture as "basic organizational principles, norms, commitments and values related to how safety and health work and their relative importance relative to other workplace goals" [125]. This is almost consistent with the obligation of management [124]. In other words, the obligation of management is to create a good enterprise safety culture, so as to enhance safety atmosphere Wait for the level. Zhou et al. [31] believed that management's safety commitment is the core of safety climate, and good safety commitment helps to improve the safety climate and improve workers' behavior. Mario and Mara [32] believe that the management should start with human behavior and formulate a series of reasonable safety rules and regulations, so as to create a good safety climate. He et al. [33] verified by SEM that safety climate is positively correlated with safety compliance and safety participation behavior, and good safety compliance behavior and workers' safety participation contribute to the improvement of safety climate. Fang et al. [38] also considered that workers' safety participation is an important part of a good safety climate.

Three additional papers are as follows:

[123]D.Zohar.Safety Climate in Industrial Organizations Theoretical and Applied Implications.J.Appli.Psycho.1980,65(1),96-102,doi:10.1037//0021-9010.65.1.96.

[124]Nicole Dedobbeleer.,François Béland.A safety climate measure for construction sites.J.Saf.Res.1991,22(2),97-103,doi:10.1016/0022-4375(91)90017-P.

[125]Al-Bayati, A. J., Albert, A. and Ford, G. Construction Safety Culture and Climate: Satisfying the Necessity for an Industry Framework.Practice.Perio.Struct. Des .Constr.2019,24 (4), doi:10.1061/(ASCE)SC.1943-5576.0000452.

Point4:

The authors discussed individual factors on page 7, but they missed a critical factor: language and culture barriers. See the following references: 

 human factors what about language and cultural barriers? check the following: 

Al-Bayati, A. J., Abudayyeh, O., Fredericks, T., and Butt, S. (2017) "Managing Cultural Diversity at U.S. Construction Sites: Hispanic Workers' Perspectives." Journal of Construction Engineering and Management, DOI:10.1061/(ASCE)CO.1943-7862.0001359

Chan, A., Javed, A., Lyu, S., Hon, C., and Wong, F. (2016). "Strategies for improving safety and health of ethnic minority construction workers." J. Constr. Eng. Manage.,10.1061/(ASCE)CO.1943-7862.0001148,05016007.

Response4:

Thank you for your valuable suggestions. By sorting out the literature you summarized, we added a language and cultural barrier factor to the individual influencing factors. The details are as follows:

Due to the differences of regional and cultural environment, the communication between workers, especially from different countries and regions, will appear language barriers. Al-Bayati et al. [126] believed that language and cultural barriers are the main causes of communication failure, and some unclear construction will only lead to human errors, thus increasing the incidence of accidents. In addition, some language communication barriers can also lead to a certain degree of psychological pressure, which makes them unwilling to timely feed back to their foreign supervisors. Their work problems also lead to the recurrence of unsafe behaviors [126]. Chan et al. [127] in order to fill the research gap of migrant construction workers' safety knowledge in Eastern Asian cultural countries, the existing safety knowledge system is developed from the research mainly conducted in Western English leading countries to multi dialect Asian countries, and the training is conducted in their own languages, so as to help the safety communication among workers and reduce the occurrence of unsafe behaviors.

Point5:

The authors should also state in the introduction section as well as in the discussion section that "A human action that precursors a work-related accident could be a system error - not a human error." When organizational management factors (line 225) were the root cause (Al-Bayati et al. 2018). 

Al-Bayati, J., Abudayyeh, and Albert, A. (2018). “Managing Active Cultural Differences in U.S. Construction Workplaces: Perspectives from Non-Hispanic Workers." Journal of Safety Research, Elsevier. DOI: 10.1016/j.jsr.2018.05.004

Response5:

Thank you for your valuable suggestions. We have added your suggestions to the introduction and discussion of this article.

For example, Al-Bayati et al.[128] think that when organizational management factors are the root cause, the cause of the accident may be system error, not human error.

Point6:

Line 164. There is a reference missing. 

Response6:

I'm very sorry, this is a small mistake in our work. The reference of this passage is from Yang and Kim [12].

[12].Yang, Yong Koo.,Byung-Seok, Kim.Study on the Structural Relation between the Level of Fatigue and Stress of Construction Workers and Disaster Risks.J. Korea .Saf. Manag & Sci.2014,16,35-44.doi:10.12812/ksms.2014.16.3.35.

Point7:

The authors should clearly explain the differences between fatigue and pressure. For example, culture and language barriers lead to pressure; could they lead to fatigue as well? 

Response7:

Thank you for your valuable comments. The following is our explanation of this problem. Yang and Kim [12] think that the psychological stress of construction workers is more important than physical fatigue, which shows that the two are not the same concept. Fatigue is more likely to be perceived by the body, such as some pain and disease, or sleepiness caused by lack of rest, while stress is more likely to be perceived by the psychological level of some negative emotions, etc. we have made a detailed division at the level of individual factors. The suggestion that cultural and language barriers you mentioned will lead to stress is added to the language level of individual factors and cultural barriers. The details are as follows (in the aspect of individual factors, line165 and line230):

When it comes to the physiological factors of workers, The first thing that comes to mind is the physical condition of workers. Construction workers are faced with various health problems, including physical fatigue, cardiovascular and cerebrovascular diseases, muscle and bone diseases, and high pressure in life and work, which increase the occurrence of unsafe behaviors at work [12]. Based on the Construction Worker Fatigue Rating Scale (FASCW), Fang et al. [10] used the level of fatigue to measure the safety performance of workersand found that workers make more mistakes when they are fatigued. Girija et al. [11] found that construction workers are the group with the largest number of smokers when compared with those from other sectors. It is not only easy to cause lung infection, cough and health problems, but also easily distracts during smoking and leads to accidents.

Due to the differences of regional and cultural environment, the communication between workers, especially from different countries and regions, will appear language barriers. Al-Bayati et al. [126] believed that language and cultural barriers are the main causes of communication failure, and some unclear construction will only lead to human errors, thus increasing the incidence of accidents. In addition, some language communication barriers can also lead to a certain degree of psychological pressure, which makes them unwilling to timely feed back to their foreign supervisors. Their work problems also lead to the recurrence of unsafe behaviors [126]. Chan et al. [127] in order to fill the research gap of migrant construction workers' safety knowledge in Eastern Asian cultural countries, the existing safety knowledge system is developed from the research mainly conducted in Western English leading countries to multi dialect Asian countries, and the training is conducted in their own languages, so as to help the safety communication among workers and reduce the occurrence of unsafe behaviors.

Once again, we would like to express our sincere thanks for your review and suggestions on the manuscript. Your suggestions have greatly helped us and we have learned a lot about safety. Thank you for your suggestions.

Yours sincerely,

Zhen Li

Name:Zhen Li

E-mail:janeli@ujs.edu.cn

Cc.

Qingfeng Meng,Wenyao Liu,Xin Hu

Round 2

Reviewer 3 Report

I believe the revised manuscript is very much improved. The only think that I would like to see to to Al-Bayati (2019) cultural awareness training for migrant construction workers after the following statement ( line 238 -242): Chan et al. [127] in order to fill the research gap of migrant construction workers' safety knowledge in Eastern Asian cultural countries, the existing safety knowledge system is developed from the research mainly conducted in Western English leading countries to multi dialect Asian countries, and the training is conducted in their own languages, so as to help the safety communication among workers and reduce the occurrence of unsafe behaviors. Al-Bayati, A. J. (2019) “Satisfying the Need for Diversity Training for Hispanic Construction Workers and Their Supervisors at U.S. Construction Workplaces: A Case Study." Journal of Construction Engineering and Management, 145 (6), DOI: 10.1061/(ASCE)CO.1943-7862.0001663

Author Response

Response to Reviewer 3 Comments

Thank you for your valuable comments. Your suggestions will help us to improve the manuscript better. According to your suggestion, we have made the following modifications:

Point:

I believe the revised manuscript is very much improved. The only think that I would like to see to to Al-Bayati (2019) cultural awareness training for migrant construction workers after the following statement ( line 238 -242): Chan et al. [127] in order to fill the research gap of migrant construction workers' safety knowledge in Eastern Asian cultural countries, the existing safety knowledge system is developed from the research mainly conducted in Western English leading countries to multi dialect Asian countries, and the training is conducted in their own languages, so as to help the safety communication among workers and reduce the occurrence of unsafe behaviors.

 Al-Bayati, A. J. (2019) “Satisfying the Need for Diversity Training for Hispanic Construction Workers and Their Supervisors at U.S. Construction Workplaces: A Case Study." Journal of Construction Engineering and Management, 145 (6), DOI: 10.1061/(ASCE)CO.1943-7862.0001663 .

Response:

According to your suggestion, we summarize the articles of Al Bayati and add them to the manuscript(Line 243-247).Some improvements have been made to this paragraph. The details are as follows:

Due to regional and cultural differences, communication between workers (especially from different countries and regions) will have language barriers. Al-Bayati et al. [126] believe that language and cultural barriers are the main sources of communication failure, and some unclear construction instructions will lead to human error and increase the incidence of accidents. In addition, some language communication barriers can also cause a certain degree of psychological pressure. This communication pressure makes some workers unwilling to report their work problems to their foreign supervisors in time, which also leads to some unsafe behaviors repeatedly [126]. In subsequent research, Al-Bayati et al.[136] added the unit of "cultural diversity training" to the safety training module on construction sites in the United States. After providing Hispanic workers with Spanish scene training, workers’ safety capabilities has been greatly improved. This case also proves the importance of language and cultural communication.Chan et al. [127] also suggested that the construction industry in developed countries should try to train migrant workers in their native language when training them; at the same time, they also encourage migrant workers to learn the local language. Through these methods, it is helpful for safe communication between workers and interaction between workers and supervisors, and the incidence of unsafe behaviors will also be greatly reduced.

Once again, we would like to express our sincere thanks for your review and suggestions on the manuscript.

Yours sincerely,

Zhen Li

Name:Zhen Li

E-mail:janeli@ujs.edu.cn

Cc.

Qingfeng Meng,Wenyao Liu,Xin Hu
